# Mitochondrial Fission Governed by Drp1 Regulates Exogenous Fatty Acid Usage and Storage in Hela Cells

**DOI:** 10.3390/metabo11050322

**Published:** 2021-05-18

**Authors:** Jae-Eun Song, Tiago C. Alves, Bernardo Stutz, Matija Šestan-Peša, Nicole Kilian, Sungho Jin, Sabrina Diano, Richard G. Kibbey, Tamas L. Horvath

**Affiliations:** 1Yale Program in Integrative Cell Signaling and Neurobiology of Metabolism, Department of Comparative Medicine, Yale University School of Medicine, New Haven, CT 06511, USA; jaeeun.song@yale.edu (J.-E.S.); bernardo.stutz@yale.edu (B.S.); matija.sestan-pesa@yale.edu (M.Š.-P.); 2Laboratory Medicine, Institute for Clinical Chemistry, Technische Universität Dresden, 01069 Dresden, Germany; tiago.alves@uniklinikum-dresden.de; 3Centre for Infectious Diseases, Parasitology, Heidelberg University Hospital, 69120 Heidelberg, Germany; nicole.kilian@med.uni-heidelberg.de; 4Institute of Human Nutrition, Columbia University Irving Medical Center, New York, NY 10032, USA; sj3094@cumc.columbia.edu (S.J.); sd3449@cumc.columbia.edu (S.D.); 5Department of Molecular Pharmacology and Therapeutics, Columbia University Irving Medical Center, New York, NY 10032, USA; 6Department of Internal Medicine, Yale University School of Medicine, New Haven, CT 06511, USA; Richard.kibbey@yale.edu; 7Department of Neuroscience, Yale University School of Medicine, New Haven, CT 06511, USA; 8Department of Ob/Gyn & Reproductive Sciences, Yale University School of Medicine, New Haven, CT 06511, USA

**Keywords:** mitochondrial dynamics, fatty acid oxidation, lipid homeostasis

## Abstract

In the presence of high abundance of exogenous fatty acids, cells either store fatty acids in lipid droplets or oxidize them in mitochondria. In this study, we aimed to explore a novel and direct role of mitochondrial fission in lipid homeostasis in HeLa cells. We observed the association between mitochondrial morphology and lipid droplet accumulation in response to high exogenous fatty acids. We inhibited mitochondrial fission by silencing dynamin-related protein 1(DRP1) and observed the shift in fatty acid storage-usage balance. Inhibition of mitochondrial fission resulted in an increase in fatty acid content of lipid droplets and a decrease in mitochondrial fatty acid oxidation. Next, we overexpressed carnitine palmitoyltransferase-1 (CPT1), a key mitochondrial protein in fatty acid oxidation, to further examine the relationship between mitochondrial fatty acid usage and mitochondrial morphology. Mitochondrial fission plays a role in distributing exogenous fatty acids. CPT1A controlled the respiratory rate of mitochondrial fatty acid oxidation but did not cause a shift in the distribution of fatty acids between mitochondria and lipid droplets. Our data reveals a novel function for mitochondrial fission in balancing exogenous fatty acids between usage and storage, assigning a role for mitochondrial dynamics in control of intracellular fuel utilization and partitioning.

## 1. Introduction

Mitochondrial dynamics, fusion and fission of mitochondria, control mitochondrial morphology, which can reflect on different cell types [1,2], cell cycle [3] and nutrient and toxic stress the cell might be facing [4,5,6]. The main emphasis of mitochondrial dynamics has been on the quality control of the mitochondrial population, which is heavily interconnected with cellular bioenergetics such as starvation or nutrient excess, because mitochondria are central organelles of metabolism [4,7]. Interestingly, the relationship between mitochondrial dynamics and lipid metabolism is somewhat contradicting. When cells become more reliant on fatty acids under starvation, elongated mitochondria are observed [5,8,9]. On the other hand, nutrient excess with high concentration of palmitic acids, the most abundant saturated fatty acids in the human body, induces mitochondrial fragmentation [5,10]. While studies have shown the importance of mitochondrial fusion in lipid metabolism during starvation via rescuing damaged mitochondria, there is a shortage of studies exploring the role of mitochondrial fission in lipid metabolism because mitochondrial fragmentation has been often regarded as a phenotype of damaged mitochondria caused by lipotoxicity [5,9]. The mechanistic connection between mitochondrial fission and lipid metabolism was first suggested via adenosine monophosphate (AMP)-activated protein kinase (AMPK), a major enzyme in metabolic homeostasis. Classically, AMPK has been shown to regulate fatty acid metabolism through inactivation of acetyl-CoA carboxylase (ACC), enabling carnitine palmitoyltransferase-1 (CPT1) to translocate long-chain fatty acids into mitochondria for oxidation [11,12]. More recently, mitochondrial fission factor (MFF) was identified as a substrate for AMPK, showing AMPK can directly regulate mitochondrial fission by mediating the recruitment of dynamin-related protein 1 (DRP1) to the mitochondrial outer membrane through MFF activation, proposing a link between mitochondrial fission and lipid metabolism [13].

Supporting the aforementioned findings, we have previously shown that hypothalamic Agouti-related protein (AgRP) neuronal activity is associated with smaller and more numerous mitochondria in these cells [14,15]. We also found that these changes in mitochondrial morphology in AgRP neurons paralleled the induction of a intracellular long chain fatty acid utilizing pathway by the mitochondria involving AMPK and CPT1 [15]. Subsequently, we unmasked that mitochondrial dynamics, including mitochondrial fusion and fission, impact the activity of AgRP and nearby pro-opiomelanocortin (POMC) neurons in a fuel availability-dependent manner [16,17]. These observations raised the hypothesis that mitochondrial fission and fusion processes are tightly connected to fuel availability and that these processes are defining to how fuels are used within cells [6]. Specifically, our data indicated that mitochondrial fission is an inherent element in proper utilization of long chain fatty acids by cells. We also showed that DRP1 knockout in AgRP neurons lowered mitochondrial fatty acid oxidation, impacting AgRP activities and body weight [18]. To address this more directly at cellular level, we used HeLa cells in this study to investigate the association between mitochondrial fission and fatty acid utilization.

Metabolic homeostasis is about achieving energetic balance through the usage and storage of an energy source. For lipid metabolism, the main players are mitochondria for fatty acid oxidation and lipid droplets for the storage of fatty acids as triacylglycerols [19,20]. Rather than simply responding to conditions of nutrient starvation or excess, mitochondrial dynamics can direct the lipid usage-storage balance within the cell, thus revealing a more complex relationship between mitochondrial dynamics and the state of bioenergetics. Here, we demonstrate in HeLa cells that palmitic acid propagates mitochondrial fission, which, in turn, is crucial for mitochondrial uptake and metabolism of fatty acids to maintain lipid homeostasis.

## 2. Results

### 2.1. Mitochondrial Morphology Reflects the Exogenous Fatty Acid Usage and Storage

Mitochondria are the major sites for fatty acid oxidation to generate cellular energy, whereas lipid droplets store free fatty acids as triacylglycerols. To gain a better understanding of the relationship between mitochondrial dynamics and fatty acid metabolism, we analyzed mitochondrial morphology and the amount of lipid droplets in HeLa cells incubated with different fatty acids. Cells incubated in base medium (BM) which contains only glucose and glutamine showed elongated mitochondria. Interestingly, oleic acid (OA) and palmitic acid (PA) induced opposite outcomes in mitochondrial morphology and the amount of lipid droplets in cells (Figure 1A). To examine the changes in mitochondrial morphology caused by OA or PA, we measured different mitochondrial parameters in the cells. Cells incubated in BM + PA had more, smaller mitochondria with more circular/shorter morphology, suggesting higher mitochondrial fragmentation or fission while BM + OA caused no change in any mitochondria parameters compared to mitochondria in cells incubated with no exogenous fatty acid (Appendix A). To quantify these mitochondrial parameters, we divided the total number of mitochondria by total area of mitochondria in a cell (Figure 1B). Cells incubated in BM + OA contained elongated mitochondria with increased accumulation of lipid droplets, whereas cells incubated in BM + PA contained smaller, less elongated mitochondria indicative of fission or fragmentation and no change in the amount of lipid droplets (Figure 1C,D). Since mitochondria and lipid droplets work together in lipid usage/storage balance, we wanted to determine whether the amount of lipid droplets was indicative of fatty acid oxidization by mitochondria. Mitochondrial respiration linked to ATP production was significantly increased when cells were incubated in BM + PA while cells incubated in BM + OA did not show increased ATP production, indicating that PA was more readily oxidized by mitochondria while OA was incorporated into lipid droplets (Figure 1E).

To further investigate the connection between mitochondrial fission/fragmentation and PA oxidation, we first confirmed the presence of PA was sufficient to induce mitochondrial fission/fragmentation. When cells were incubated in glucose only (Glc) or glucose with PA (Glc + PA), mitochondria in cells incubated with Glc + PA were less tubular and smaller which is indicative of increased fission/fragmentation (Appendix A). To confirm this, additional PA was used in mitochondria as an energy source via fatty acid oxidation. We then compared the oxygen consumption rates (OCR) of cells incubated in different combinations of nutrients and the contribution of fatty acids to the TCA cycle using ^13^C-glucose to measure the amount of glucose used in the TCA cycle. Fatty acids and glucose are both converted into acetyl-CoA, which is a major entry point into the TCA cycle, thus, the actual contribution of fatty acids to the TCA cycle can be calculated based on the amount of ^13^C-glucose that was converted into acetyl-CoA, as previously described [21]. The presence of PA increased OCR due to an increase in fatty acid usage in TCA cycle (Appendix A). The increase in fatty acid oxidation was not due to a change in mitochondrial mass measured with MitoTracker Green signal (Appendix A). Notably, cells incubated with Glc + Gln presented with elongated mitochondria and similar OCRs relative to those incubated with Glc + PA. These observations indicate that mitochondrial respiration, per se, is not related to mitochondrial morphology (Appendix A). On the other hand, a clearer correlation was found between mitochondrial morphology and fatty acid usage in the TCA cycle, in which cells with higher fatty acid usage in the TCA cycle also showed higher mitochondrial fission/fragmentation (Appendix A), These data unmasked that mitochondrial dynamics are driven by substrate usage independent of respiratory capacity of mitochondria in cells.

### 2.2. Mitochondrial Fission Directs the Distribution of Exogenous PA between Mitochondria and Lipid Droplets

Mitochondrial fission requires the recruitment of cytosolic dynamin-related protein 1 (DRP1) to the outermembrane of mitochondria. After 1-h incubation, the presence of PA increased the expression level of *DRP1* and also the colocalization of endogenous DRP1 with TOM20, a mitochondrial outermembrane protein, supporting that the changes in mitochondrial morphology in BM + PA is driven by DRP1-regulated mitochondrial fission (Figure 2A–C). To investigate the role of mitochondrial fission in fatty acid metabolism, we inhibited mitochondrial fission by silencing DRP1 and observed the effect on mitochondrial morphology and metabolic response to exogenous PA.

Cells transfected with DRP1 siRNA effectively prevented the PA-induced mitochondrial fission (Figure 2D,E). Similar to cells incubated in BM + OA (Figure 1A), DRP1-silenced cells also had significantly higher accumulation of lipid droplets in response to exogenous PA (Figure 2D,F), suggesting that the mitochondrial fission affects the distribution of exogenous fatty acids within the cell. Without PA, there was no difference in the amount of lipid droplets between DRP1-silenced cells and the control cells (Appendix A). However, after a 4-h incubation in BM + PA, we recorded reduced mitochondrial mass in DRP1 knockdown (DRP1 KD) cells, suggesting that the defect in mitochondrial fission could influence mitochondria beyond mitochondrial morphology, possibly because of the lipotoxicity, as the cellular lipid homeostasis cannot be maintained without mitochondrial fission to process high concentrations of exogenous PA in a 4-h incubation (Appendix A). We decided to observe the effect of silencing DRP1 on the earlier stage of fatty acid distribution to reduce the PA effect on mitochondrial mass in DRP1 KD cells.

To track the behavior of exogenous fatty acids within a cell, we used BODIPY 558/568 Red C12 (C12), a saturated long-chain (18-carbon) fatty acid analog with a fluorophore attached that has been used to study the movement of long-chain fatty acid in multiple studies [8,22]. C12 was mixed with PA in a 1:2000 ratio and added to BM to visualize the incorporation of exogenous fatty acids within the cell. After 1-h incubation, mitochondria and lipid droplets were labeled and fluorescence signals were visualized by spinning-disc microscopy (Figure 3B). The total C12 signal, the C12 signal from mitochondria and the C12 signal from lipid droplets from the same cell were measured with raw integrated density and the percentages of C12 signals coming from two organelles were calculated (C12-Mito% or C12-LD%) to measure the distribution of PA between mitochondria and lipid droplets (Figure 3A). To visualize the distribution of PA between mitochondria and lipid droplets, Mito-C12% and LD-C12% were plotted on xy-graph and a simple linear regression analysis was performed to find the best-fitting line. The xy-plot of C12-Mito% and C12-LD% visualized the balance between storage and usage—cells with higher C12-Mito% have lower C12-LD% and vice versa. Silencing DRP1 shifted this balance towards storage. There was a visible shift in population density towards high C12-LD% and low C12-Mito% when DRP1 was silenced (Figure 3C). The percentage of C12 signals from mitochondria was decreased and C12 signal incorporated into lipid droplets was significantly increased in DRP1 KD cells (Figure 3D,E). In DRP1 KD cells, lipid droplet biogenesis also seemed to be increased in response to PA. Specifically, the addition of PA significantly increased the mRNA level of diacylglycerol acyltransferase 1 (*DGAT1*) in DRP1 KD cells (Appendix A). Interestingly, the increase in DGAT1-dependent-lipid droplet biogenesis, along with highly fused mitochondria, was observed in starved cells to avoid lipotoxicity caused by free fatty acids released from autophagic breakdown of organelles [23]. These phenotypes of mitochondria and lipid droplets in starved cells resemble DRP1 KD cells incubated in BM + PA, suggesting that mitochondrial fission could be involved in determining how cells respond to high cellular lipid environments.

AMP-activated protein kinase (AMPK) is a major metabolic enzyme involved in lipid metabolism through ACC-CPT1 and mitochondrial fission through MFF-DRP1 (Figure 3F). AMPK can be activated by elevated AMP level during starvation or long-chain fatty acyl-CoA. How these two nutrient challenges induce different changes in mitochondrial morphology is unclear, however, we first wanted to confirm that PA-induced mitochondrial fission is linked to AMPK-regulated lipid metabolism [12,13,24,25]. To determine this, we observed the metabolic effect of AMPK activation using 5-aminoimidazole-4-carboxamide ribonucleotide (AICAR) on DRP1 KD cells (Figure 3F,G). AICAR caused a significant increase in the ATP-linked OCR only when PA was present. However, this increase in OCR induced by AICAR disappeared when DRP1 was silenced, strongly implying that AMPK can regulate long-chain fatty acid oxidation not only through ACC but also through mitochondrial fission (Figure 3G).

Along with mitochondrial fission, DRP1 is involved in other organelles behaviors, such as ER-mitochondria interactions and peroxisomal fission [26,27]. We did not observe significant differences in ER-mitochondria contact between control and DRP1 KD cells in both BM and BM + PA incubation (Appendix A). Peroxisomes are where very long chain fatty acid breakdown occurs [26]. Although mitochondria are the primary location of fatty acid oxidation, fatty acids with more than 22 carbons are initially processed by peroxisomes. To assess whether the PA-oxidation difference in DRP1 KD we observed is related to peroxisomal oxidation, we measured the expression of levels of peroxisomal oxidation enzymes—fatty acyl-CoA oxidase (ACOX1), 3-ketoacyl-CoA thiolase (ACAA1) and peroxisomal bifunctional enzyme (EHHADH). The presence of PA increased the expressions of these genes in both control and DRP1 KD cells with no significant difference and there was no difference in non-mitochondrial oxygen consumption (Appendix A). DRP1 KD cells had the lower OCR related to mitochondrial fatty acid oxidation, measured by using etomoxir, a CPT1 inhibitor, further confirming that the metabolic effect of DRP1 we observed is specifically related to mitochondrial fatty acid oxidation (Figure 3H). Interestingly, the incubation in BM + PA increased the expression of CPT1A in control cells but DRP1 KD suppressed this increase (Figure 3I). Since CPT1A is also regulated by AMPK, we decided to investigate the effect of CPT1A expression level on mitochondrial morphology and fatty acid metabolism.

### 2.3. Mitochondrial Fission Works Independently of CPT1A on Fatty Acid Distribution

CPT1 is a mitochondrial outer membrane enzyme responsible for transferring the acyl group of long-chain fatty acyl-CoA to carnitine, which is then translocated into the mitochondria [28,29]. It regulates the rate-limiting step in long-chain fatty acid oxidation and is part of AMPK-linked lipid metabolism. We first observed the changes in mitochondrial morphology of cells with different expression levels of CPT1A. Both the overexpression and knockdown of CPT1A increased the mitochondrial fragmentation in BM and BM + PA (Figure 4A,B and Appendix A). The addition of PA still increased mitochondrial fragmentation in both CPT1A mutants, suggesting that PA-induced mitochondrial fission occurs independent of CPT1 expression level (Figure 4B and Appendix A). There was also no difference in the amount of lipid droplets in cells with CPT1A-overexpression and CPT1A-knockdown compared to their corresponding control cells (Appendix A).

To observe and compare the effect of CPT1A on the distribution of PA with mitochondrial fission, we used CPT1A overexpression (CPT1A OE) cells and tracked the C12 distribution between mitochondria and lipid droplets after 1-h incubation using C12. CPT1A OE significantly lowered C12-LD% without affecting the C12-Mito% (Figure 4D,E), even though mitochondria in CPT1A OE cells had the higher fatty acid oxidation-linked OCR response to PA (Figure 4C). The xy-plot of C12-Mito% and C12-LD% of individual cells showed that the cells with similar C12-Mito% had lower C12-LD% when CPT1A is overexpressed (Figure 4F). Unlike DRP1 KD cells, which showed the populational shift towards higher C12-LD% and lower C12-Mito%, CPT1A OE cells showed the similar distribution of population as their control cells but with lower C12-LD%. In addition, CPT1A OE lost its effect on C12-LD% after 4 h incubation, while DRP1 KD cells maintained the shift in balance, suggesting that CPT1A OE did not cause a shift in the cellular balance of usage/storage (Appendix A). Together, these data suggest that CPT1A OE cells increased the rate of fatty acid processing in mitochondria, leaving less PA to be incorporated into lipid droplets.

To further explore other bioenergetic parameters more comprehensively, we assessed the impact of CPT1A OE by ^13^C-MIMOSA precursor-product isotopomer analysis using mass spectroscopy to measure the metabolic flux of glycolysis and the TCA cycle [21,30]. To our surprise, CPT1A OE cells had significantly higher cataplerosis (the removal of metabolites from the TCA cycle) through phosphoenolpyruvate carboxykinase (PEPCK) and/or malic enzyme compared to PA alone or CPT1A overexpression alone (Figure 4G). These experiments were performed at a metabolic steady state (4-h incubation), so the amount of cataplerosis must be balanced by an equivalent amount of anaplerosis (the entrance of metabolites into the TCA cycle) [21,31]. While CPT1A OE increased the flux of TCA cycle when exposed to PA, it had no effect on the contribution of fatty acid in TCA cycle and the ATP-linked mitochondrial respiration, supporting that CPT1A OE did not change the balance of fatty acid usage and storage (Appendix A). These data all together suggest that CPT1A OE increased the fatty acid processing capacity of mitochondria but did not change the fatty acid usage/storage balance as observed in DRP1 KD cells.

PA induced the expression level of both DRP1 and CPT1A (Figure 2C and Figure 3I). These two enzymes both play critical but separate roles in lipid homeostasis. Mitochondrial fission regulates the balance of exogenous fatty acid distribution between mitochondria and lipid droplets and CPT1A determines the mitochondrial capacity to process fatty acids. The disruption in the CPT1A expression level in DRP1 KD cells that we observed could be resulted from the shift in the balance of fatty acid distribution towards storage when PA induced-mitochondrial fission is inhibited.

## 3. Discussion

In this study, we examined the function of mitochondrial fission in lipid homeostasis. In a high-lipid environment, an excessive amount of fatty acid can be either stored in lipid droplets or used in mitochondria to reduce potential lipotoxicity and to maintain metabolic homeostasis [23,32,33]. We demonstrated that DRP1-mediated mitochondrial fission was directly involved in the balance between fatty acid storage and usage by facilitating fatty acid uptake by mitochondria.

Our findings show that cellular response to higher lipid usage by the mitochondria involves mitochondrial dynamics and is a flexible system able to accommodate different situations. Our observations showed the connection between the balance of lipid usage/storage and mitochondrial morphology. Both starvation and incubation with OA increased the amount of lipid droplets and mitochondrial elongation [8,23]. While incubation with PA caused no increase in lipid droplets, it did promote mitochondrial usage of fatty acids and mitochondrial fission. Although the mechanism is uncertain, we tried to clarify the role of mitochondrial fission in cellular response to high amount of intracellular fatty acids. We approached this question from the aspect of AMPK signaling, which connects mitochondrial fission and mitochondrial long-chain fatty acid oxidation. In addition, AMPK can be activated via the increased level of intracellular AMP under starvation and via exogenous long-chain fatty acyl-CoA, making a good target to understand the role of mitochondrial fission in lipid metabolism [24,34].

The activation of AMPK with AICAR increased the ATP-linked OCR only when PA was present and this increase disappeared when DRP1 was silenced, implying the effect of mitochondrial fission on exogenous fatty acid oxidation. The PA-induced mitochondrial fission was regulated by DRP1, independent of CPT1 expression level. Both CPT1A OE and CPT1A KD still increased mitochondrial fragmentation in response to PA while DRP1 KD successfully inhibited PA-induced changes in mitochondrial morphology. While we were able to confirm that CPT1A controls the mitochondrial capacity to process fatty acids, we also defined its limitation. The balance of fatty acid usage and storage was strongly influenced by DRP1-regulated mitochondrial fission. DRP1 KD cells showed elongated mitochondria and increases in lipid droplets in the presence of exogenous PA, similar to cells under starvation. This observation implies the difference in cellular responses to high lipid environments when induced by starvation and exogenous PA is related to mitochondrial fission. AMPK can be activated by (1) increased intracellular AMP levels under starvation and (2) long-chain fatty acyl-CoA, metabolically active form of long-chain fatty acids [24]. However, starvation does not promote mitochondrial fission even though we observed that mitochondrial fission is involved in AMPK-mediated lipid metabolism. A potential explanation is that the mitochondrial fusion prevails in starvation due to its protective effect on mitochondria from non-selective autophagosomal degradation that occurs during starvation [9,35]. This shows the complex and flexible nature of lipid homeostasis and the diverse roles of mitochondrial dynamics in lipid metabolism. Mitochondrial fission could be the key player in deciding how cells response to high fatty acids in the cytoplasm, as we were able to mimic the cellular phenotypes of starvation in DRP1 KD cells. In this study, we demonstrated that PA-induced mitochondrial fission was directly and obligatorily involved in lipid homeostasis by facilitating the mitochondrial uptake of fatty acids, thereby regulating the balance between fatty acid usage by mitochondria and storage by lipid droplets. These findings give direct cellular biological support to the notion that mitochondrial dynamics, specifically mitochondrial fission controlled by long chain fatty acids, may drive cellular lipid homeostasis.

## 4. Materials and Methods

### 4.1. Cell Culture

ATCC HeLa CCL2 (ATCC) were cultured at 37 °C and 5% CO_2_ in DMEM (Gibco 11965092) supplemented with 10% fetal bovine serum (Gibco, Grand Island, NY, USA) and penicillin/streptomycin (Gibco, Grand Island, NY, USA). For experiments, cells were incubated in Live Cell Imaging Solution (Invitrogen) for 2 h prior to incubation in experimental nutrient combinations. The base medium was prepared with DMEM (Gibco A1443001), 20 mM HEPES, and 10% charcoal-stripped fetal bovine serum (Gibco, Grand Island, NY, USA) and supplemented with different nutrient combinations. PA and OA were dissolved in ethanol to make 100 mM stock solutions. Once different combinations of nutrients were added to the base medium, the media were warmed up to 37 °C for at least one hour before it was administered to the cells.

A CPT1A-overexpressing stable HeLa cell line was grown under the same culture conditions as WT HeLa cells, except with the addition of 1 mg/mL geneticin (Gibco, Grand Island, NY, USA).

### 4.2. Transfection

CPT1A plasmid with neomycin selection marker (A1436) was purchased from GeneCopoeia (Rockville, MD, USA). Plasmid transfections were performed with Lipofectamine 2000 (ThermoFisher, Carlsbad, CA, USA) as recommended by the manufacturer. Single cell sorting was performed to select for cell colonies with the highest level of CPT1A overexpression as confirmed by western blotting (Appendix A).

CPT1A siRNA (AM16708) and its negative control Non-coding siRNA (AM4611) were purchased from ThermoFisher. DRP1 siRNA (SI02661365) and its negative control Non-coding siRNA (SI03650325) were purchased from QIAGEN (Cambridge, MA, USA). The siRNA transfections were performed with Lipofectamine RNAiMAX (Invitrogen, Carlsbad, CA, USA). Prior to experiments, cells were incubated with 40 nM siRNA and 4 μL for CPT1A KD and 5 μL for DRP1 KD RNAiMAX in OptiMEM (Gibco, Grand Island, NY, USA) for 4–5 h, and then incubated with growth medium for 48 h. For the Seahorse experiment with siRNA transfection, the same conditions were used, but the reverse transfection protocol from the manufacturer was performed to minimize the media change.

### 4.3. Imaging

Cells were counted and seeded on a glass-bottom dish (3.5-cm diameter, No. 1.5 MatTek, Ashland, MA, USA) coated with fibronectin (Millipore) 1 day before imaging as previously described [36]. Mitochondria were labeled with 100 nM MitoTracker Green FM/Orange CMTMRos/Deep Red FM (Invitrogen, Eugene, OR, USA) for 3 min, and then washed and incubated in Live Cell Imaging Solution at 37 °C for 10 min prior to imaging. Lipid droplets were labeled with HCS LipidTOX Green Neutral Lipid Stain (Invitrogen) as recommended by the manufacturer. Furthermore, 100 mM BODIPY 558/568 C12 (C12, Invitrogen, Eugene, OR, USA) was prepared, mixed with 100 mM PA at a 1:2000 ratio and then added to BM instead of PA for visualization of PA incorporation into mitochondria and lipid droplets. All pictures were taken with live cells using spinning disk confocal microscopy (SDCM).

Colocalization of DRP1 and TOM20 z-stack images are taken with Zeiss LSM510. Cells were fixed in 4% paraformaldehyde for 20 min, washed with PBS, permeabilized with 0.3% NP40, 0.05% Triton-X100 in PBS for 3 min and incubated with corresponding primary antibodies to TOM20 and DRP1 overnight at 4 °C followed by Alexa 488- and Alexa 647-labelled secondary antibodies next day for 60 min at room temperature.

### 4.4. Transmission Electron Microscopy

Cultured cells were fixed in 2.5% glutaraldehyde in 0.1 M sodium cacodylate buffer (pH 7.4) at room temperature. After cells were rinsed in the same buffer twice, they were further post-fixed in 1% OsO_4_ in 0.1 M cacodylate buffer at room temperature for one hour. Samples were further stained en bloc with 2% aqueous uranyl acetate for 30 min, subsequently dehydrated in a graded series of ethanol to 100% and embedded in Embed 812 resin. Blocks were polymerized in 60 °C oven overnight. Thin sections (60 nm) were cut by a Leica UC7 ultramicrotome and post-stained with 2% uranyl acetate and lead citrate. Sample sections were examined with a FEI Tecnai transmission electron microscope at accelerating voltage of 80 kV. Digital images were recorded with an Olympus Morada CCD camera and iTEM imaging software.

### 4.5. RT-qPCR

RNA was extracted using QIAGEN Rneasy Micro Kit (#74004). cDNA was synthetized using QIAGEN Whole Transcriptome Kit (#207043). The gene expression was investigated using real-time qPCR, performed in 96 or 384-well PCR plates using the Roche 480 LightCycler Thermal Cycler (Roche, Indianapolis, Indiana). The real-time PCR reaction mixture contained i-Taq SYBR green master mix (BioRad), 0.6 mM primer pairs, and diluted cDNA in a total volume of 10 µL The mixture was heated initially to 95 °C for 3 min to activate hot-start iTaq DNA polymerase and then followed by 50 cycles with denaturation at 95 °C for 10 s, annealing at 60 °C for 45 s, and extension at 72 °C for 60 s. Samples and standards were run in triplicate. Primers (Table 1) were carefully designed using NCBI Primer-BLAST (https://www.ncbi.nlm.nih.gov/tools/primer-blast/; accessed on 1 July 2020). Gene and mRNA sequences were obtained from NCBI (http://www.ncbi.nlm.nih.gov/Tools/; accessed on 1 July 2020). Primers were tested for efficiency, specificity and primer dimer formation using 10-fold dilution curve of cDNA concentration. Additionally, a melt curve protocol designed for increment temperatures of 0.5 °C with a starting temperature of 57 °C and ending at 92 °C was performed at the end of all PCR-reactions. Threshold cycle values (Ct) and relative expression of genes were evaluated using the comparative threshold cycle method (ΔΔCt) using a reference gene (actin).

### 4.6. Mitochondrial Mass and ROS Measurement Using Flow Cytometry

Cells were treated with experimental conditions prior to labeling with MitoTracker Green or MitoSOX Red (Invitrogen, Eugene, OR, USA) for 3 min at 37 °C for measurements of mitochondrial mass and ROS prior to analysis by flow cytometry. FlowJo software was used to calculate the mean of the control sample that was used to normalize the values of experimental conditions.

### 4.7. Metabolism Assay

The OCR was measured using a 96-well XFe extracellular flux analyzer (Seahorse Bioscience). Cells were seeded at a density of 2 × 10^4^ per well. For OCR measurements during 4-h incubations, the cells were incubated in Live Cell Imaging Solution for 2 h, then in experimental nutrient conditions for 3 h before switching to the same experimental conditions prepared in Seahorse Media. The cells were incubated in Seahorse Media for 1 h in non-CO_2_ 37 °C incubator.

For measurement of fatty acid oxidation change in response to PA: [(OCR value with BSA-PA)−(OCR value after etomoxir injection)]−[(OCR value with BSA)−(OCR value after etomoxir injection)].

For calculations of ATP-linked OCR, basal OCR and maximal OCR, we followed the manufacturer instruction in XF cell Mito Stress Test user guide available on their website.

For AICAR experiment, cells were first incubated in Live Cell Imaging Solution with AICAR for 2 h then incubated in 5 mM Glc + 5 mM Gln + BSA or 5 mM Glc + 5 mM Gln + 100 μM BSA-conjugated PA in Seahorse Media for 1 h before the experiment.

The following drugs were used: oligomycin (1 μM), FCCP (2 μM), rotenone + antimycin A (0.25 μM each) (Seahorse Bioscience, Santa Clara, CA, USA), etomoxir (50 μM), AICAR (500 μM) (Sigma). ^13^C-glucose experiment was performed and analyzed as previously published [21].

### 4.8. Image Analysis

Mitochondrial area and number as well as the area of lipid droplets were quantified with the analyze particle function on Fiji. All C12 fluorescence signals were measured with raw integrated density. Background signals of C12 were measured for individual picture and subtract it from C12 measurements. Mitochondria and lipid droplets ROIs were selected using analyze particle and C12 signals were measured from selected ROIs.

Colocalization analysis was done as previously published [13]. Analysis was done by using colocalization plugin in ImageJ to highlight the colocalization points in white pixels. The colocalization points were divided by total mitochondrial area in each z-stack pictures and then normalized to the values of BM group in each experiment.

For mitochondria-ER contact quantification, the numbers of mitochondria-ER contact were scored and divided by corresponding cytoplasm area measured in high-magnification electron microscope images.

### 4.9. Statistical Analyses

Statistical analyses were performed with Prism 8 (GraphPad). Simple linear regression was performed in Prism 8. Comparisons for two groups were calculated using unpaired two-tailed Student’s *t*-tests, one-way ANOVA followed by Bonferroni’s multiple comparison tests for more than two groups, and two-way ANOVA for experiments with more than two variants followed by multiple comparison.

## Figures and Tables

**Figure 1 metabolites-11-00322-f001:**
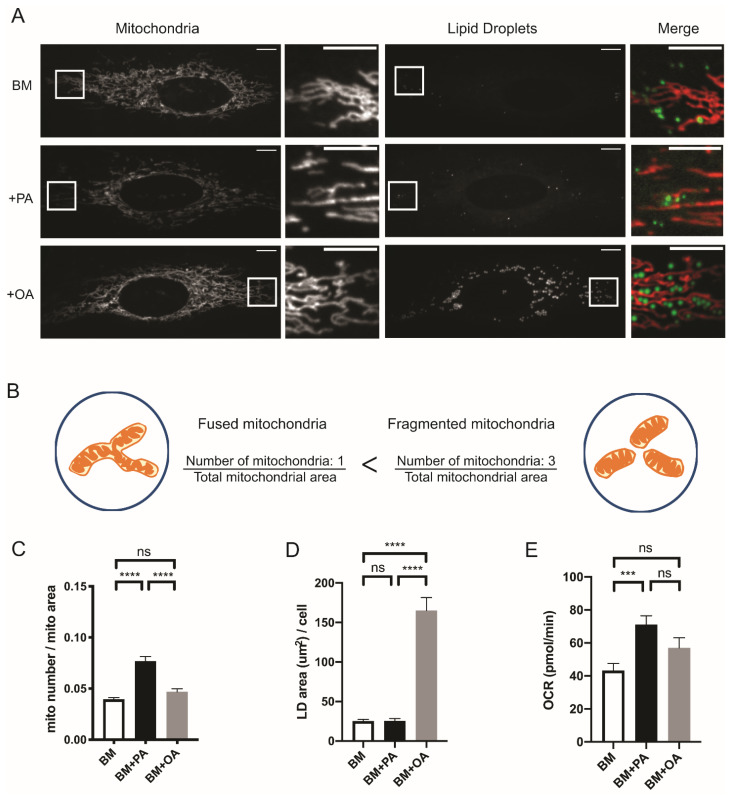
Mitochondrial morphology is associated with the exogenous fatty acid usage and storage. (**A**) Representative images of live HeLa cells incubated in 5 mM glucose + 5 mM glutamine (BM); 5 mM glucose + 5 mM glutamine + 100 μM palmitic acid (BM + PA); 5 mM glucose + 5 mM glutamine + 100 μM oleic acid (BM + OA) for 4 h. Mitochondria are labeled with MitoTracker Orange and lipid droplets are labeled with LipidTOX Green. Scale bars = 10 μm. (**B**) Cartoon of mitochondria in a cell demonstrating how mitochondrial morphology was calculated. The total number of mitochondria was divided by the total area of mitochondria per cell to quantify the fragmentation of mitochondria. (**C**) Quantification of mitochondrial morphology. BM (*n* = 47); BM + PA (*n* = 45); BM + OA (*n* = 44). 3 independent experiments; Data are expressed as mean ± S.E.M. Ordinary one-way ANOVA-Tukey’s multiple comparisons test. (**D**) Quantification of Lipid Droplets contents. BM (*n* = 43); BM + PA (*n* = 41); BM + OA (*n* = 46). 3 independent experiments. Data are expressed as mean ± S.E.M. Ordinary one-way ANOVA-Tukey’s multiple comparisons test. (**E**) Mitochondrial oxygen consumption rate (OCR) associated with ATP respiration. BM (*n* = 27); BM + PA (*n* = 25); BM + OA (*n* = 24). 3 independent experiments. Data are expressed as mean ± S.E.M. Ordinary one-way ANOVA-Tukey’s multiple comparisons test. ns = not significant; *** *p* < 0.001; **** *p* < 0.0001. All pictures were taken with spinning-disc confocal microscopy.

**Figure 2 metabolites-11-00322-f002:**
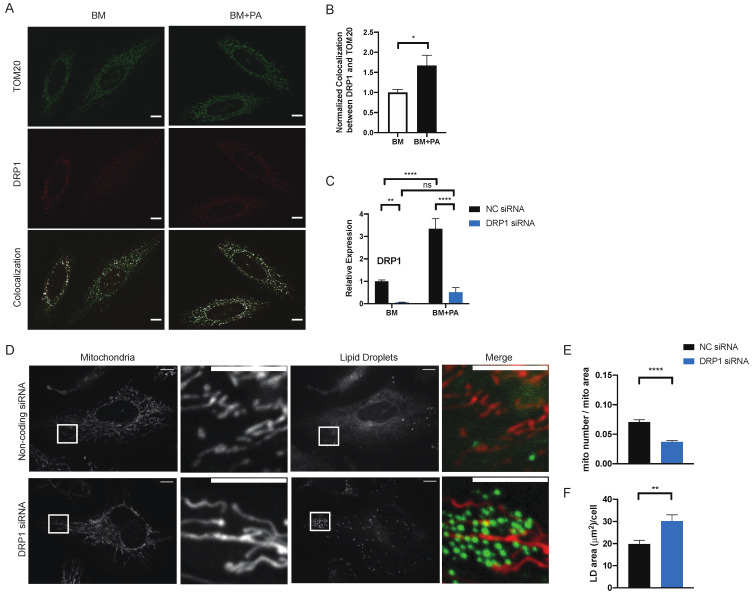
PA-induced mitochondrial fission is regulated by DRP1. (**A**) Representative images of HeLa cells incubated in BM or BM + PA for 1-h prior and then fixed and labelled with TOM20 and DRP1 antibodies. Colocalization of TOM20 and DRP1 was identified by colocalization plugin (ImageJ). Scale bars = 10 μM. (**B**) Quantification of (**A**). The values were normalized to BM. *n* = 13 z-stack images. ~40 cells. 3 independent experiments. Data are expressed as mean ± S.E.M. Unpaired two-tailed *t*-test. * *p* < 0.05 (**C**) Normalized expression level of DRP1 using RT-qPCR. *n* = 5. Data are expressed as mean ± S.E.M. 2 way ANOVA-Sidak’s multiple comparison test. ** *p* < 0.01 **** *p* < 0.0001. (**D**) Representative images of live HeLa cells after transfections with non-coding siRNA (NC siRNA) and DRP1 siRNA after 4-h incubation with BM + PA. Mitochondria are labelled with MitoTracker DeepRed and lipid droplets are labelled with LipidTOX Green. Scale bars = 10 μm. Quantification of mitochondrial morphology (**E**) and lipid droplets (**F**) of cells shown in (**A**). NC siRNA (*n* = 42); DRP1 siRNA (*n* = 44). 3 independent experiments. Data are expressed as mean ± S.D. Unpaired two-tailed *t*-test. **** *p* < 0.0001; ** *p* < 0.01.

**Figure 3 metabolites-11-00322-f003:**
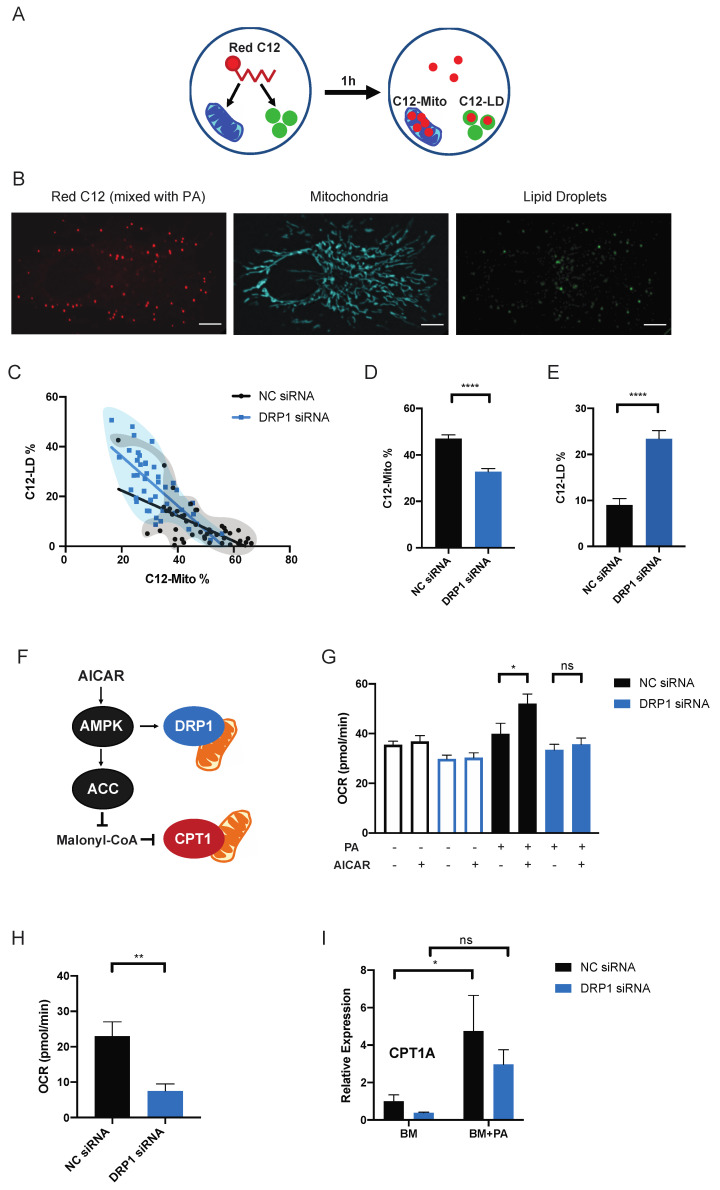
DRP1 directs the distribution of exogenous fatty acid between mitochondria and lipid droplets. (**A**) After 1-h incubation with Red C12, the total Red C12 fluorescence intensity per cell and the C12 fluorescence intensity coming from mitochondrial ROI (C12-Mito) and lipid droplets ROI (C12-LD) were measured. C12-Mito or C12-LD was divided by the corresponding total C12 to calculate the percentages of C12 fluorescence intensity coming from mitochondria and C12 fluorescence intensity coming from lipid droplets per cell. (**B**) Representative images of live HeLa cells incubated with Red C12 (100 mM C12 was mixed with 100 mM PA in 1:2000 ratio) and labeled with MitoTracker DeepRed and LipidTOX Green. Scale bars = 10 μm. (**C**) The xy-plot of C12-Mito% and C12-LD% of cells transfected with NC siRNA or DRP1 siRNA to visualize the distribution of fatty acids between mitochondria and lipid droplets within an individual cell. The measurements of individual cells are plotted. The populations of control cells (black) and DRP1 KD cells (blue) are marked with translucent bubbles to visualize the populational shift. Simple linear regression values: NC siRNA: slope = −0.5797, R^2^ = 0.4264; DRP1 siRNA: slope = −0.9942, R^2^ = 0.5573. Percentage of C12 signal coming from mitochondria (**D**) and from lipid droplets (**E**) calculated within individual cells. *n* = 45. 3 independent experiments. Data are expressed as mean ± S.E.M. Unpaired two-tailed *t*-test. **** *p* < 0.0001. (**F**) AMPK pathway connecting MFF-DRP1 mitochondrial fission and ACC-CPT1 fatty acid oxidation. (**G**) The effect of AICAR on mitochondrial ATP-linked OCR. Cells were incubated with or without 500 µM AICAR for 2 h then in BM or BM + PA for 1 h in Seahorse media. -: absence; +: presence. *n* = 12–15. Data are expressed as mean ± S.E.M. Unpaired two-tailed *t*-test * *p* < 0.05. (**H**) Measurement of fatty acid oxidation in response to PA using etomoxir. NC siRNA (*n* = 7) DRP1 siRNA (*n* = 9). Data are expressed as mean ± S.E.M. Unpaired two-tailed *t*-test ** *p* < 0.01. (**I**) Normalized expression levels of *CPT1A* using RT-qPCR. *n* = 5. Data are expressed as mean ± S.E.M. 2 way ANOVA-Sidak’s multiple comparison test. * *p* < 0.05.

**Figure 4 metabolites-11-00322-f004:**
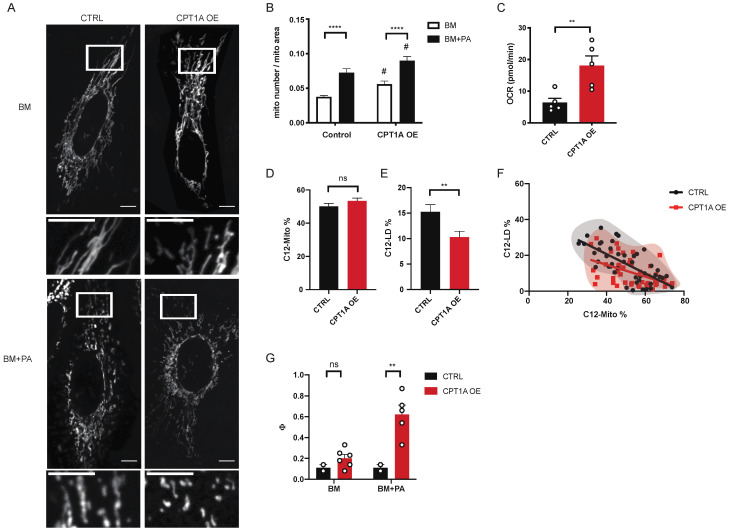
CPT1A regulates the mitochondrial capacity of fatty acid processing. (**A**) Representative images of live WT HeLa cells (CTRL) and CPT1A overexpression HeLa cells (CPT1A OE) incubated in BM or BM + PA. Scale bars = 10 μm. (**B**) The quantification of mitochondrial morphology. CTRL: BM (*n* = 33), BM + PA (*n* = 29); CPT1A OE: BM (*n* = 33), BM + PA (*n* = 31). 3 independent experiments. Data are expressed as mean ± S.E.M. 2 way ANOVA-Sidak’s multiple comparison test. **** *p* < 0.0001 (BM vs BM + PA); # *p* < 0.05 (CTRL vs CPT1A OE). (**C**) Fatty acid oxidation-linked OCR response to PA using etomoxir. Data are expressed as mean ± S.E.M. Unpaired two-tailed *t*-test. ** *p* < 0.01. (**D**) The percentage of Red C12 fluorescence intensity coming from mitochondria (C12-Mito%) and (**E**) The percentage of Red C12 fluorescence intensity coming from lipid droplets (C12-LD%). *n* = 46. 3 independent experiments. Data are expressed as mean ± S.E.M. Unpaired two-tailed *t*-test. ** *p* < 0.01. (**F**) The xy-plot of C12-Mito% and C12-LD% to visualize the distribution of fatty acids between mitochondria and lipid droplets within an individual cell. The measurements of individual cells are plotted. The populations of control cells (black) and CPT1A OE cells (red) are marked with translucent bubbles to visualize the populational shift. Simple linear regression values: CTRL: slope = −0.5386, R^2^ = 0.4784. CPT1A OE: slope = −0.3328, R^2^ = 0.2251. (**G**) Measurement of cataplerotic reactions (Φ: the ratio of malate to pyruvate positional enrichment) of the TCA cycle using 13C MIMOSA precursor-product isotopomer analysis. Data are expressed as mean ± S.E.M. 2 way ANOVA-Sidak’s multiple comparison test. ** *p* < 0.01.

**Table 1 metabolites-11-00322-t001:** Primers used

Gene Name.	Forward Primer	Reverse Primer
Actin	TGACGTGGACATCCGCAAAG	CTGGAAGGTGGACAGCGAGG
ACAA1	GACAGGTCATCACGCTGCTCAA	CCAGGGTATTCAAAGACGGCAG
ACOX1	GGCGCATACATGAAGGAGACCT	AGGTGAAAGCCTTCAGTCCAGC
CPT1A	AAATTACGTGAGCGACTGGTG	TGCTGCCTGAATGTGAGTTG
DGAT1	GCTTCAGCAACTACCGTGGCAT	CCTTCAGGAACAGAGAAACCACC
DRP1	GATGCCATAGTTGAAGTGGTGAC	CCACAAGCATCAGCAAAGTCTGG
EHHADH	CGGAGCATCGTGGAAAACAGCA	CCGAGTCTACAGCAATCACAGG
tRNA-Leu	CACCCAAGAACAGGGTTTGT	TGGCCATGGGTATGTTGTTA

## Data Availability

The data presented in this study are available on request from the corresponding author.

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
