# Peer review of "Mitochondrial Fission Governed by Drp1 Regulates Exogenous Fatty Acid Usage and Storage in Hela Cells"

_metabolites, 2021, doi:10.3390/metabo11050322_

Round 1

Reviewer 1 Report

The paper by Song and colleagues investigates the role of mitochondrial fission in lipid homeostasis. The authors selectively inactivated mitochondrial fission by inhibiting DRP1 and found that it caused an imbalance between fatty acid storage and usage by facilitating fatty acid uptake by mitochondria. The authors then went further mechanistically and reported that while CPT1A controlled the respiratory rate of mitochondrial fatty acid oxidation, it did not cause a shift in the distribution of fatty acids between mitochondria and lipid droplets.

This is a very well-written paper that addresses an important research question that has been largely unexplored. The authors use appropriate and well-performed approaches to address this question. I do not have any major concerns.

The only minor issue that I have is related to HeLa cells. Although this an excellent cell system to study mitochondrial morphology and lipid homeostasis, there is no mention in the Abstract that all the work was performed in this cell line. This information should be added to the Abstract. In addition, based on previous studies from the authors’ lab examining the importance of mitochondrial fusion and fission in hypothalamic neurons, it is not clear why the authors chose to study HeLa cells and not, for example, GT1-7 or mHypoE-N45 cells, which are functionally closer to hypothalamic neurons. The rationale for studying HeLa cells should be described at the beginning of the Results section.

Author Response

  1. The only minor issue that I have is related to HeLa cells. Although this an excellent cell system to study mitochondrial morphology and lipid homeostasis, there is no mention in the Abstract that all the work was performed in this cell line. This information should be added to the Abstract. In addition, based on previous studies from the authors’ lab examining the importance of mitochondrial fusion and fission in hypothalamic neurons, it is not clear why the authors chose to study HeLa cells and not, for example, GT1-7 or mHypoE-N45 cells, which are functionally closer to hypothalamic neurons. The rationale for studying HeLa cells should be described at the beginning of the Results section.

We would like to thank the reviewers for the careful review of our manuscript and for the suggestions that helped improved the manuscript.

We changed the title to “Mitochondrial fission governed by Drp1 regulates exogenous fatty acid usage and storage in Hela Cells” and edited the abstract and the introduction. We decided to use HeLa cells for its wealth of past usage in live-cell imaging and its high efficiency in transfections and staining methods. To determine its physiological effect on organs/organism, different cells lines along with in vivo model should be used in future studies.

We removed Figure 4H because we suspect an experimental error. Max respiration for CTRL is shown in Figure S2E.

Reviewer 2 Report

Concerns:

  1. Why HeLa cells? Justify use of this type of cell versus a non-cancer model.
    1. Muscle, liver, etc?
    2. How does this Drp1 proposed mechanism translate to mitochondrial fusion/energetics within other tissues?
  2. Respiration measures in a state of low demand/high energy input, any measures of ROS and/or oxidative damage? (or antioxidant systems)
    1. Justify how the low energy demand would affect lipid fate.
  3. Any measures of oxidative phosphorylation efficiency? (ATP)
    1. Reserve capacity or max capacity changes?
  4. Would blocking fatty acid entry and processing through the peroxisome effect the outcomes? Both FAs used can be processed by the peroxisome first (partially).
    1. Any lipid droplet measures only from the peroxisomal contribution?
  5. Peroxisomal data/number by PMP70 or some membrane marker or activity assay?
    1. Does no change in mRNA equal no change in function or peroxisomal number?
    2. Did you measure changes in peroxisomal genes in the CPT1- OE?
      1. Fatty acids from peroxisome can enter independent of CPT1 in some tissue types and overload system/effect lipid fate.

Author Response

Please sea the attachment

Reviewer 3 Report

Mitochondrial fission governed by Drp1 regulates exogenous fatty acid usage and storage

In this study, the author analysed the association between mitochondrial morphology and lipid droplet accumulation in response to high exogenous fatty acids. The Author inhibited the mitochondrial fission by silencing dynamin-related protein 1 (DRP1) and they observed a new function for mitochondrial fission in balancing exogenous fatty acids between usage and storage.

In addition, the role of carnitine palmitoyltransferase-1 (CPT1) was studied and it was shown that (CPT1) is involved in controlling the respiratory rate of mitochondrial fatty acid oxidation but did not cause a shift in the distribution of fatty acids between mitochondria and lipid droplets.

It is an interesting aspect to find out that, mitochondrial fission can play a role in fatty acid homeostasis and to find out that excessive fission is not always associated with functional defect that lead to human diseases.

Testing the hypothesis with only one cell line (Hela cells) minimizes the importance of the results. The conclusion that mitochondrial fission maintain fatty acid homeostasis can only be stated for Hela cells. We still do not know if this result represent a general mitochondrial behavior.  

  • All figures are too small and all images have a bad quality. The mitochondrial morphological changes are not comprehensible from the images.

  • Line 103-106 and figure 1E as well as line 297-301 and Figure 4H, S5D

In this part ATP production was linked from oxygen consumption rate (OCR). From method and material, it was not clear how metabolism in XFe extracellular flux anlayzer was performed. With “Mito Stress Kit” for example it is able to calculate direct ATP produced and the maximal respiration as well spare respiratory capacity. Changes in maximal respiration is an indication of different substrate intake or changes in the biogenesis of the mitochondria. A graph of ATP and graph of maximal respiration gives further information about mitochondrial metabolism in Hela cells in the presence of different substrates.

  • In Line 155-161 cells transfected with DRP1 siRNA are morphologically similar to cells with BM + OA and had significantly higher accumulation of lipid droplets. In contrast, in “Supplemental Figure 3” it was stated that no differences in mitochondrial morphology or amount of lipid droplets were observed between cells transfected with NC siRNA and DRP1 siRNA. Clear this contradiction.

  • In this study, only mitochondria from Hela cells were analysed. It is not known if mitochondrial morphological change in different nutrients state is a general occurrence or a specification for Hela cell. The Title should be changed to 

“Mitochondrial fission governed by Drp1 regulates exogenous fatty acid usage and storage in Hela Cells”.

  • From the results, it was demonstrated that mitochondrial fragmentation is caused only after incubation with excess of palmitic acid. However, with oleic acid no morphological changes could be detected. How do you explain this result?

Round 2

Reviewer 3 Report

Why did the authors insert the MitoSox Red from Thermofisher to materials and methods in 4.6. I didn´t find any place in the manuscript where results about ROS were shown.